# Analysis of the Protective Effects of *Rosa roxburghii*-Fermented Juice on Lipopolysaccharide-Induced Acute Lung Injury in Mice through Network Pharmacology and Metabolomics

**DOI:** 10.3390/nu16091376

**Published:** 2024-04-30

**Authors:** Zhiyu Chen, Shuo Zhang, Xiaodong Sun, Duo Meng, Chencen Lai, Min Zhang, Pengjiao Wang, Xuncai Huang, Xiuli Gao

**Affiliations:** 1State Key Laboratory of Functions and Applications of Medicinal Plants, School of Pharmaceutical Sciences, Guizhou Medical University, Guiyang 550025, China; chenzhiyu@stu.gmc.edu.cn (Z.C.); zhangshuo@gmc.edu.cn (S.Z.); sunxiaodong@gmc.edu.cn (X.S.); 18296002014@163.com (D.M.); 2022010030128@stu.gmc.edu.cn (C.L.); minzhang@gmc.edu.cn (M.Z.); wangpengjiao@gmc.edu.cn (P.W.); hz8863549@163.com (X.H.); 2Center of Microbiology and Biochemical Pharmaceutical Engineering, Guizhou Medical University, Guiyang 550025, China; 3Experimental Animal Center of Guizhou Medical University, Guiyang 550025, China; 4Guizhou Provincial Engineering Research Center of Food Nutrition and Health, Guizhou Medical University, Guiyang 550025, China

**Keywords:** *Rosa roxburghii*-fermented juice, metabolomics, network pharmacology, molecular docking

## Abstract

Acute lung injury, a fatal condition characterized by a high mortality rate, necessitates urgent exploration of treatment modalities. Utilizing UHPLS-Q-Exactive Orbitrap/MS, our study scrutinized the active constituents present in *Rosa roxburghii*-fermented juice (RRFJ) while also assessing its protective efficacy against LPS-induced ALI in mice through lung histopathological analysis, cytokine profiling, and oxidative stress assessment. The protective mechanism of RRFJ against ALI in mice was elucidated utilizing metabolomics, network pharmacology, and molecular docking methodologies. Our experimental findings demonstrate that RRFJ markedly ameliorates pathological injuries in ALI-afflicted mice, mitigates systemic inflammation and oxidative stress, enhances energy metabolism, and restores dysregulated amino acid and arachidonic acid metabolic pathways. This study indicates that RRFJ can serve as a functional food for adjuvant treatment of ALI.

## 1. Introduction

Acute lung injury (ALI) is a severe illness with a high death rate. It is characterized by inflammation, damage to the endothelium and epithelial barriers, and problems with gas exchange [1,2]. ALI can lead to the excessive or long-term activation of immune cells in the body, followed by the release of destructive pro-inflammatory and proapoptotic mediators, disrupting the balance of inflammation/anti-inflammatory, oxidation/reduction, and cell apoptosis, leading to lung pathology [3,4,5]. The development of ALI and Acute Respiratory Distress Syndrome (ARDS) is closely related to the secretion of inflammatory factors. ALI/ARDS has multiple potential causes, including viral infection, severe trauma, and blood transfusion [6,7]. Lipopolysaccharides (LPS) are surface glycolipids produced by Gram-negative bacteria, which can cause severe inflammatory reactions in the lungs. ALI mice induced by LPS exhibit similar pathological damage to COVID-19 [8,9]. Although we have a good understanding of the pathogenesis of ALI/ARDS and have made progress in clinical treatment methods for ALI/ARDS, the quality of life of patients has been severely reduced by adverse reactions caused by treatment. Therefore, exploring treatment methods for ALI/ARDS remains crucial [10].

*Rosa roxburghii* Tratt (RRT) is primarily found in the southwestern parts of China, specifically Yunnan and Guizhou. RRT is a wild resource that belongs to the Rosaceae family. It serves as both a medicinal plant and a functional food [11,12]. RRT contains a wide range of nutrients, such as vitamins, sugars, carbohydrates, organic acids, proteins, amino acids, dietary fiber, trace elements, and other nutrients. It also contains various active ingredients, including SOD, flavonoids, polyphenols, polysaccharides, triterpenes, sterols, glycosides, and volatile components. RRT exhibits antioxidant properties, modulates the immune system, lowers blood sugar levels, regulates glucose and lipid metabolism, influences intestinal flora and lipid metabolism, prevents the formation of atherosclerosis, and performs other tasks [13,14,15,16]. Despite RRT’s extensive track record of being a safe and traditional health food, it cannot be stored for more than forty days without proper control over the storage conditions. Fermentation is not only regarded as a highly effective method for preserving food, but it is also known to convert edible raw materials into new bioactive chemicals that possess distinct immunological, blood sugar, and anti-inflammatory capabilities [17]. Hence, investigating the mechanisms of RRFJ could be significant in expanding the range of therapeutic options for acute lung injury (ALI).

Metabolomics involves monitoring the comprehensive and dynamic alterations in metabolites that occur during drug treatment. This process aids in the diagnosis of diseases, the identification of treatment targets, and the detection of disease-specific biomarkers. Additionally, it offers solutions for preventing and monitoring the effectiveness and safety of drugs [18,19,20]. Network pharmacology is a developing field that investigates the processes of diseases and the effects of drugs by considering biological networks. Through the examination of the interplay among disease, gene, protein, and drug, this study elucidates the intricate connections between biological systems, drugs, and diseases from a network standpoint. Consequently, it unveils the intricate relationship between traditional Chinese medicine and diseases [21,22,23]. Molecular docking is a computer-based technique used to forecast the interactions between molecules and biological targets. Its purpose is to uncover compounds that could potentially have significant implications for human health. The typical procedure involves initially predicting the molecular orientation of the ligand in the recipient body, followed by assessing their complementarity using an evaluation function [24,25].

Hence, this study integrates metabolomics and network pharmacology approaches to elucidate RRFJ’s effect on ALI and its potential mechanisms. First, untargeted metabolomics was used to evaluate the effect of RRFJ on ALI to discover potential metabolites and related metabolic pathways. Network pharmacology was subsequently employed to predict the targets and associated pathways of the active components in RRFJ for the treatment of ALI. The objective was to determine the main mechanism of RRFJ in treating ALI. Additionally, molecular docking studies were conducted to anticipate the interactions between key compounds and their predicted targets. This investigation contributes to a deeper understanding of RRFJ’s therapeutic principles in ALI prevention and treatment.

## 2. Materials and Methods

### 2.1. Reagents

The *Rosa roxburghii*-fermented juice was provided by Guizhou Shanwangguo Health Industry Co., Ltd., a company based in Guizhou, China. The lipopolysaccharide (LPS) from Escherichia coli serotype 055:B5 was acquired from Sigma-Aldrich. The Dexamethasone sodium phosphate injection (DEX) was obtained from Suicheng Pharmaceutical Co., Ltd. in Zheng-zhou, China. It has a concentration of 5 mg/mL and the batch number is 22,009,281. The ELISA kits for mice TNF-α, IL-10, IL-6, and IL-1β were obtained from Shenzhen NeoBioscience Co., Ltd. in Shenzhen, China. All reagents and chemicals, except for UPLC-grade reagents (formic acid, acetonitrile, and methanol from Merck), were of analytical quality.

### 2.2. UHPLC-ESI-Q-Exactive Plus Orbitrap-MS Analysis

To initially detect potentially active chemicals in RRFJ, we employed a Thermo Vanquish Horizon UHPLC system that was equipped with Hypersil Gold C18 columns (2.1 × 100 mm^2^, 1.9 μm). The optimized gradients consisted of phase A, which was a 0.1% aqueous solution of formic acid, and phase B, which was a 0.1% acetonitrile solution of formic acid. The concentration of B in the solution changes with time as follows: from 0 to 3 min, it is 2%; from 3 to 23 min, it increases to 98%; from 23 to 26 min, it remains at 98%; from 26 to 26.1 min, it decreases to 2%; and from 26.1 to 28 min, it remains at 2%. The injection volume, flow rate, and column temperature were adjusted to 2 μL, 0.3 mL/min, and 40 °C, respectively. Following that, the ESI-Q-Exactive Plus Orbitrap system was utilized for mass spectrometry analysis, covering both negative and positive ion modes. The configuration of the mass spectrum parameters was as follows. The spray voltage was set to 3.5/2.5 kV with a tolerance of +/−, the carburetor temperature was 350 °C, and the capillary temperature was 320 °C. The conventional fullms–ddms method was used, with a scanning range of 100–1500 *m*/*z*. The MS1 resolution was 70,000, while the MS/MS resolution was 17,500. The step normalized collision energies (NCE) were set to 20, 40, and 60. Raw data imported into Compound Discoverer 3.2 were expected to have a relative molecular mass deviation of less than 5 ppm. The investigation utilized accurate relative molecular mass and primary and secondary mass spectrometric fragmentation data obtained from Xcalibur software (Xcalibur 4.3, Thermo Fisher Scientific, Waltham, MA, USA; sourced from Guizhou Medical University, Guiyang, China), in addition to standard spectrum and fragmentation information from *m*/*z* Cloud, *m*/*z* Vault, Masslist, MoNA, ChemSpider databases, and relevant literature. In order to enhance the dependability of the results, the advanced recognition algorithm mzLogic was utilized. Furthermore, Mass Frontier 7.0 software (Thermo Fisher Scientific, Waltham, MA, USA; sourced from Guizhou Medical University, Guiyang, China) was utilized to perform fragment-assisted inference and validation in order to determine the active chemicals present in the RRFJ.

### 2.3. Animals Experiments

A total of 50 specific pathogen-free (SPF) male BALB/C mice, weighing 18–22 g, were procured from the Animal Centre of Guizhou Medical University (License no. SCXK (xiang) 2022-0011) (Changsha Tianqin Biotechnology Co., Ltd., Changsha, China). These mice were housed in standard laboratory conditions with a 12/12 h light/dark cycle, temperature maintained at 25 ± 1 °C, and humidity at 60 ± 5%, with ad libitum access to water and standard rodent diet. The experimental procedures were conducted in strict compliance with the “Guidelines for the Care and Use of Experimental Animals” and were approved by the Ethics Committee of Guizhou Medical University (license no.SYXK (qian) 2018-0001). After a one-week acclimatization period, 50 mice with temperature fluctuations less than 0.5 °C were selected to commence the experiment. Subsequently, these 50 mice were randomly divided into five groups: control group, LPS group, Dexamethasone (DEX) group, low-dose treatment group (LPS + RRFJ 10 mL/kg), and high-dose treatment group (LPS + RRFJ 20 mL/kg). The ALI model was induced by intratracheal instillation of 2 mg/kg LPS. The control group and model group received 10mL/kg of physiological saline for a week. The DEX group received 10 mL/kg saline for the first four days and 2 mg/kg dexamethasone via intraperitoneal injection for the subsequent three days. The RRFJ group received RRFJ of different concentrations via gavage for a week. Mice in the model group and RRFJ group were sacrificed 24 h after LPS administration. The plasma was obtained using centrifugation at a speed of 3500 revolutions per minute for a duration of 10 min at a temperature of 4 °C. It was then stored at a temperature of −80 °C for subsequent analysis. The right lung’s superior lobe was promptly rinsed with cold phosphate-buffered saline (PBS) and preserved in 10% paraformaldehyde for histopathological analysis utilizing hematoxylin and eosin (H&E) staining. Lung tissue was homogenized and centrifuged for 10 min at 4000 rpm, and the supernatant was collected to prepare a 10% tissue homogenate.

### 2.4. Lung Histopathology

The lung tissues of mice were preserved in 4% paraformaldehyde, dried out, enclosed in paraffin, cut into 4 μm slices, and dyed with hematoxylin and eosin (H&E). Afterwards, the sections were examined using a light microscope.

### 2.5. Measurement of Lung Wet/Dry Ratio

After rinsing with cold PBS, the wet weight of the right lower lobe of the lung was measured, followed by drying in a 60 °C oven for 72 h before obtaining the dry weight. The wet-to-dry (W/D) ratio was calculated using the formula (wet weight/dry weight) × 100%.

### 2.6. Determination of Cytokine Concentration

The levels of tumor necrosis factor-α (TNF-α), interleukin-6 (IL-6), interleukin-1β (IL-1β), and interleukin-10 (IL-10) in plasma and lung tissue were assessed using ELISA kits following the manufacturer’s instructions.

### 2.7. Determination of Lung MDA, SOD, GSH, and MPO

Malondialdehyde (MDA) content, Superoxide Dismutase (SOD) activity, Glutathione (GSH) content, and Myelopeoxidase (MPO) content were determined using assay kits from Nanjing Jiancheng Bioengineering Institute, following the manufacturer’s instructions.

### 2.8. Plasma and Lung Sample Preparation 

A 100 μL plasma sample was mixed with 300 μL of a solution containing equal parts acetonitrile and methanol (volume/volume). Subsequently, the mixture was subjected to incubation at a temperature of −20 °C for a duration of 30 min, followed by centrifugation at a speed of 14,000 revolutions per minute at a temperature of 4 °C for a duration of 15 min. The resultant liquid portion was gathered for subsequent examination. A 100 mg sample of lung tissue was extracted using 0.5 mL of a pre-cooled solution of methanol and acetonitrile in water (2:2:1, *v*/*v*/*v*). The mixture was made uniform and subjected to vortex-sonication for 15 min at a low temperature. To cause proteins to separate from the mixture, the mixtures were allowed to sit overnight at a temperature of −20 °C and then subjected to centrifugation for a duration of 15 min at a speed of 15,000 revolutions per minute. The supernatants obtained were gathered for analysis. Ultimately, the liquid portion of both plasma and lung tissue samples underwent filtration using a 0.22 μm membrane. The resulting filtered liquid was then injected into a sample vial for the purpose of conducting metabolomics analysis.

### 2.9. Metabolomics Analysis

Furthermore, to conduct metabolomics analysis, we employed a UHPLC-HESI-Q-Exactive Plus Orbitrap-MS instrument, which was equipped with a ZORBAX Eclipse Plus C18 chromatography column (2.1 × 100 mm^2^, 1.8 μm; Agilent Technologies, Santa Clara, CA, USA). The optimized gradient consisted of water (0.1% formic acid, A) and acetonitrile (0.1% formic acid, B), with the following gradient: 0–2.5 min, 2–2% B; 2.5–5 min, 2–40% B; 5–12 min, 40–100% B; 12–16 min, 100–100% B; 16–16.1 min, 100–2% B; 16.1–19 min, 2–2% B. The Compound Discoverer 3.2 was employed to preprocess mass spectrometry data, which involved identifying peaks, matching peaks, and correcting retention time. The software used for conducting OPLS-DA and PCA analysis was SIMCA-P 14.1 (Umetrics, AB, Malmo, Sweden). Endogenous metabolites were identified by comparing primary and secondary mass spectrometry fragmentation information using the KEGG and HMDB databases. Furthermore, the composition was verified using MS2 spectroscopy with the internal metabolite fragment spectral library. MetabolAnalyst 5.0 facilitated the analysis of metabolic impacts associated with ALI after RRFJ prophylactic therapy, with metabolic pathways having an influence value >0.1 considered significant.

### 2.10. Network Pharmacology Analysis

The active compounds were obtained from the literature and the Traditional Chinese medicine and chemical component database (http://www.chemcpd.csdb.cn, accessed on 10 March 2024). Their structures were collected from the PubChem database (https://pubchem.ncbi.nlm.nih.gov/, accessed on 10 March 2024).The obtained compound structure was identified as potential targets for RRFJ active ingredients through searching SwissTargetPrediction (www.swisstargetprediction.ch/, accessed on 10 March 2024), The Encyclopedia of Traditional Chinese Medicine database (www.tcmip.cn/ETCM/index.php/Home/Index/index.html, accessed on 10 March 2024), and the TCMSP (the Traditional Chinese Medicine Systems Pharmacology database and analysis platform) (https://tcmspw.com/tcmsp.p-hp, accessed on 10 March 2024). ALI-related targets were searched from GeneCards (https://www.genecards.org/, accessed on 10 March 2024), OMIM (Online Mendelian Inheritance in Man) (https://omim.org/, accessed on 10 March 2024), and DrugBank database (https://drugbank.com/, accessed on 10 March 2024) using “Acute Lung Injury” as the keyword, setting the species type to “Homo sapiens”, obtaining the target name and removing duplicates, and establishing a disease target library. The active compound targets and disease targets were intersected to yield a common target for ALI and active compounds. The common targets were imported to STRING (https://cn.string-db.org/, accessed on 10 March 2024) online platform for analysis, and the results were imported to the Cytoscape 3.9.1 platform for network topology process analysis and core target screening. Key targets were annotated using the DAVID platform for GO function and KEGG enrichment analysis (*p* < 0.01), the species type was limited to “Homo sapiens”, and the results were visualized. The “Compound-target-pathway” relationship network diagram of ALI and the active compounds was obtained by using the Cytoscape 3.9.1 software platform, where nodes represented compounds, targets, and pathways and represented the interaction between them.

### 2.11. Comprehensive Analysis of Metabolomics and Network Pharmacology

The differential metabolites identified in metabolomics were imported into MetScape 3.1.3 to construct a “Metabolite-Reaction-Enzyme-Gene” network. The related genes of metabolites were then combined with the target genes and metabolic pathways screened by network pharmacology to identify key metabolic pathways and core targets.

### 2.12. Molecular Docking

The crystal structures of 10 ALI candidate protein targets were downloaded from the RCSB (The Research Collaboratory for Structural Bioinformatics protein database) at http://www.pdb.org/ accessed on 10 March 2024. Subsequently, these target proteins underwent modification using AutoDock Tool 1.5.7 software, which involved processes such as hydrogenation, dehydration, ligand removal, and amino acid optimization. Following modification, the structures were saved in pdbqt format. The three-dimensional chemical structures of the 10 active ingredients were collected from PubChem and subjected to energy minimization, with the results saved in MOL.2 format. These compounds were then imported into AutoDock Tool 1.5.7, where all flexible bonds were set to be rotatable by default, and they were saved as docking ligands in pdbqt format. Docking was performed using AutoDock Vina 1.1.2, with docking results visualized using PyMOL Molecular Graphics System, Version 3.0.

## 3. Results

### 3.1. Tentative Identification of RRFJ’s Active Ingredients

*Rosa roxburghii*-fermented juice (RRFJ) was characterized using UHPLC-ESI-Q-Exactive Plus Orbitrap-MS. A total of 44 active ingredients were preliminarily identified, including 6 amino acids, 10 organic acids, 16 flavonoids, 2 saccharide, 3 alkaloid, 1 alcohol, 2 phenolic acid, 1 phenylpropanoids, 2 terpenoids, and 1 sphingolipid. Compound 7, with a retention time of 0.993 min, yielded a precursor ion at *m*/*z* 191.0192 based on primary mass spectrometry information. The preliminary molecular formula was inferred to be C_7_H_12_O_6_. Upon collision-induced dissociation, consecutive losses of H_2_O, H_2_O, and CO_2_ fragments appeared at *m*/*z* 173.04477 and 85.02814, respectively. According to the literature, it was tentatively identified as quinic acid [26]. Compound 20, with a retention time of 2.074 min, exhibited a precursor ion at *m*/*z* 169.0136 based on primary mass spectrometry information, suggesting a preliminary molecular formula of C_7_H_6_O_5_. Upon collision-induced dissociation, consecutive losses of COOH and 2H_2_O fragments appeared at *m*/*z* 125.02316 and 79.01754, respectively. According to the literature, it was preliminarily identified as gallic acid [27]. Compound 28, with a retention time of 7.268 min, yielded a precursor ion at *m*/*z* 1290.07458 based on primary mass spectrometry information, indicating a preliminary molecular formula of C_15_H_14_O_6_. Upon collision-induced dissociation, consecutive losses of CO_2_, C_4_H_4_O_2_, and C_6_H_6_O_2_ fragments appeared at *m*/*z* 245.0448, 205.04993, and 179.03412, respectively. According to the literature, it was preliminarily identified as catechin [28]. Compound 35, with a retention time of 8.815 min, showed a precursor ion at *m*/*z* 301.0002 based on primary mass spectrometry information, suggesting a preliminary molecular formula of C_7_H_6_O. Upon collision-induced dissociation, consecutive losses of CO, CO_2_, CO_2_, and CO fragments appeared at *m*/*z* 283.99600, 257.00867, and 229.01373, respectively [29]. According to the literature, it was preliminarily identified as ellagic acid. The rich compounds in RRFJ may play a key role in preventing and treating ALI (Table 1). 

### 3.2. Effects of RRFJ on Histological Change in ALI Mice

To assess the effects of RRFJ on changes in the structure and composition of lung tissue 24 h after injecting LPS, we analyzed the morphology and histology of mouse lung tissue using hematoxylin–eosin (H&E) staining. Figure 1 shows that when normal saline is administered by the oropharyngeal route, it does not cause any major structural alterations in the lungs of healthy mice. On the other hand, animals with LPS-induced acute lung injury showed significant infiltration of inflammatory cells, substantial thickening of the alveolar wall, and increased blood flow in the lungs. Nevertheless, the administration of RRFJ and DEX effectively improved the severity of lung pathology in mice with acute lung injury (ALI). The results indicate that RRFJ has a beneficial effect in treating acute lung injury generated by LPS.

### 3.3. Effects of RRFJ on W/D Ratio in LPS-Treated Mice

The W/D ratio is a reliable measure of the severity of pulmonary edema. During this study, mice that were exposed to inhaled oropharyngeal LPS showed an increase in the W/D ratio compared to the mice who were not exposed to LPS. Significantly, the administration of RRFJ or DEX effectively prevented the rise in the W/D ratio induced by LPS, as shown in Figure 2A. These findings indicate that RRFJ and DEX can reduce the permeability of pulmonary capillaries and improve pulmonary edema in mice with acute lung injury (ALI).

### 3.4. RRFJ Inhibits Inflammatory Response in LPS-Induced ALI

Pro-inflammatory cytokines such as TNF-α, IL-6, and IL-1β, coupled with the anti-inflammatory cytokine IL-10, have important functions in the inflammatory process that leads to the advancement of acute lung injury (ALI). To assess the effect of RRFJ on LPS-induced ALI in mice, the levels of TNF-α, IL-6, IL-1β, and IL-10 in both plasma and lung tissue were measured using ELISA (Figure 2B,C). The concentrations of TNF-α, IL-6, and IL-1β in both the plasma and lung tissue of the model group were significantly elevated compared to those in the control group, although the levels of IL-10 were dramatically reduced (*p* < 0.05). Nevertheless, the prior administration of DEX and RRFJ effectively reduced the LPS-induced increase in TNF-α, IL-6, and IL-1β and the decrease in IL-10 levels in both plasma and lung tissue.

### 3.5. RRFJ Inhibits Oxidative Stress in LPS Induced ALI

Oxidative stress reactions are involved in the development of acute lung injury (ALI) and can result in the progression of pulmonary fibrosis in its later stages. SOD and GSH are enzymatic antioxidants in the body that can reduce oxidative stress. MDA can disturb biological macromolecules and initiate the release of inflammatory cytokines. However, MPO can stimulate the production of ROS and facilitate lipid peroxidation. Therefore, we evaluated the levels of GSH, SOD, MPO, and MDA in lung tissue using a reagent kit (Figure 2D–G). Our data indicated that the levels of MDA content and MPO activity in the lung tissue of mice that received pre-treatment with RRFJ and DEX were reduced compared to the model group. Conversely, the activities of SOD and GSH were significantly greater.

### 3.6. Metabolomics Analysis in Plasma and Lung Tissue

To examine the distribution of plasma and lung tissue and verify the model’s reliability, Principal Component Analysis (PCA) was employed to analyze metabolic profile data without using grouping information. The results demonstrated distinct variations among samples from distinct groups of plasma and lung tissues in mice (Figure 3A,B). Based on thresholds such as VIP > 1, P < 0.05, FC > 1.2, or FC < 0.8, 22 differential metabolites were identified in plasma, and 23 differential metabolites were identified in lung tissue. These metabolites were cross-validated using the HMDB database. To visualize the expression of differential metabolites across the sample groups, heatmap analysis was conducted. Among these metabolites, 15 were upregulated and 7 were downregulated in plasma, while 16 were upregulated and 7 were downregulated in lung tissue (Figure 3C–E). These findings indicate that RRFJ treatment effectively mitigates metabolic disorders in ALI mice. Candidate metabolites were subjected to metabolic pathway enrichment analysis using the MetaboAnalyst database. This analysis revealed 18 metabolic pathways in plasma and 17 metabolic pathways in lung tissue, including shared pathways such as phenylalanine, tyrosine, and tryptophan biosynthesis. Notably, glyoxylate and dicarboxylate metabolism, phenylalanine metabolism, arachidonic acid metabolism, glyoxylate metabolism, and dicarboxylate metabolism were significantly impacted in plasma (Figure 3F), while tryptophan metabolism, alanine, aspartate, glutamate, and tyrosine metabolism were significantly affected in lung tissue (Figure 3G).

### 3.7. GO and KEGG Enrichment and Network Analysis of Targets Related to RRFJ Activity on ALI

To investigate the mechanism underlying RRFJ’s efficacy against ALI, we conducted a network pharmacology analysis. We gathered 245 targets associated with 44 active ingredients of RRFJ from SwissTargetPrediction, the Encyclopedia of Traditional Chinese Medicine database, and the TCMSP. Subsequently, we collected 8691 ALI-related targets from GeneCards, OMIM, and the DrugBank database. After intersecting these datasets, we identified 218 potential targets for RRFJ in ALI treatment (Figure 4A). These 218 key targets were then analyzed using the STRING platform and visualized as protein–protein interaction (PPI) diagrams using Cytoscape 3.9.1 software, from which 40 core targets were selected (Figure 4B). In Figure 4G, we present the 44 active ingredients, 738 potential targets, and the 20 enriched KEGG pathways associated with the RRFJ treatment of ALI (Figure 4G). The significance of these active components was evaluated based on their centrality, intermediary, and proximity in the network. The top ten components by degree value were catechin, 4-coumaric acid, ellagic acid, abscisic acid, chrysin, emodin, gallic acid, 4-hydroxybenzaldehyde, ethyl gallate, and quercetin, which are considered key compounds for RRFJ’s treatment of LPS-induced ALI. To predict the potential mechanism of RRFJ in treating ALI, we conducted GO and KEGG enrichment analyses. In the GO enrichment analysis, biological process (BP) terms included signal transduction, protein phosphorylation, positive regulation of transcription from RNA polymerase II promoter, negative regulation of apoptotic process, response to xenobiotic stimulus, positive regulation of gene expression, positive regulation of cell proliferation, positive regulation of transcription, DNA-templated, negative regulation of transcription from RNA polymerase II promoter, and protein autophosphorylation (Figure 4D). Molecular function (MF) terms comprised protein binding, identical protein binding, ATP binding, enzyme binding, protein serine/threonine/tyrosine kinase activity, metal ion binding, protein kinase activity, protein kinase binding, zinc ion binding, and protein homodimerization activity (Figure 4E). Cellular component (CC) terms included cytosol, cytoplasm, plasma membrane, nucleus, nucleoplasm, membrane, extracellular exosome, extracellular region, mitochondrion, and integral component of plasma membrane (Figure 4F). According to KEGG enrichment analysis, 20 pathways were significantly affected, including metabolic pathways, PI3K-Akt signaling pathway, Rap1 signaling pathway, MAPK signaling pathway, Ras signaling pathway, AGE-RAGE signaling pathway in diabetic complications, HIF-1 signaling pathway, cAMP signaling pathway, chemical carcinogenesis—reactive oxygen species, apoptosis, coronavirus disease—COVID-19, TNF signaling pathway, PD-L1 expression and PD-1 checkpoint pathway in cancer, IL-17 signaling pathway, Toll-like receptor signaling pathway, VEGF signaling pathway, AMPK signaling pathway, NF-kappa B signaling pathway, mTOR signaling pathway, and JAK-STAT signaling pathway (Figure 4G).

### 3.8. Comprehensive Analysis of Metabolomics and Network Pharmacology 

We integrated metabolomics and network pharmacology to construct an interactive network. Initially, we utilized MetScape to incorporate candidate metabolites from plasma and lung tissue into a “Metabolite-Reaction-Enzyme-Gene” network (Figure 5). Furthermore, we cross-referenced genes linked to candidate metabolites with potential targets identified in network pharmacology, identifying 10 pivotal targets: HAO1, MPO, PTGS2, ALOX5, CYP2D6, CYP2C9, CYP3A4, CYP2E1, TH, and KMO. These targets influence pathways such as arachidonic acid metabolism, tryptophan metabolism, glyoxylate and dicarboxylate acid metabolism, phenylalanine metabolism, and tyrosine metabolism, which potentially contribute to inflammation and apoptosis.

### 3.9. Molecular Docking Results

For a comprehensive analysis of network pharmacology and metabolomics, molecular docking was conducted among nine key targets and the top ten chemical components with degree values in RRFJ. The stability of the protein receptor and small-molecule ligand binding relies on binding energy, where lower energy indicates a more stable binding conformation. When the binding energy is below 0, the ligand can bind freely to the receptor; a value below −5 kcal/mol suggests strong binding activity with the target protein. Figure 6A illustrates the minimum binding energy from molecular docking. Additionally, partial molecular docking results are visually depicted in Figure 6B–H. The affinity between compounds and targets predominantly falls below −5.0 kcal/mol, indicating a strong potential for close interaction between key compounds and targets. This suggests that RRFJ may exert therapeutic effects on ALI through these compounds.

## 4. Discussion

Acute lung injury is a disorder that causes inflammation throughout the body. It is defined by damage to the membrane that separates the alveoli and capillaries in the lungs, increased permeability of blood vessels, a higher number of neutrophils, and the accumulation of fluid in the lungs. Gram-negative bacteria, such as lipopolysaccharides, elicit a strong inflammatory response, altering the normal functioning of immune cells and promoting the infiltration of inflammatory cells into the lungs, therefore contributing to acute lung injury (ALI).

In our study, H&E staining revealed that RRFJ significantly alleviated LPS-induced ALI in mice by reducing inflammatory cell infiltration, thickening alveolar walls, and pulmonary congestion. Pulmonary edema, a hallmark of LPS-induced acute lung injury, was evaluated using the W/D ratio, which decreased significantly following RRFJ intervention, indicating a reduction in ALI severity. Neutrophils, macrophages, vascular endothelial cells, and alveolar epithelial cells are implicated in ALI pathogenesis, releasing pro-inflammatory factors such as IL-1β, IL-6, and TNF-α, as well as the anti-inflammatory factor IL-10 [30]. Early intervention with RRFJ led to increased levels of anti-inflammatory factors and decreased levels of pro-inflammatory factors in both plasma and lung tissue of ALI mice, suggesting the effectiveness of RRFJ in alleviating inflammation and ALI.

Apoptosis plays a crucial role in regulating inflammation within the body and significantly contributes to the development of ALI [31]. Moreover, oxidative stress is implicated in acute lung injury, leading to pulmonary fibrosis during the advanced stages of ALI. Macrophages are identified as the primary source of ROS in lung tissue, and the release of ROS and subsequent oxidative stress plays a vital role in amplifying the inflammatory response. The initiation of the mitochondrial apoptosis pathway is triggered by the oxidative stress response. At the onset of ALI, there is a rapid surge of ROS production, overwhelming the body’s antioxidant capacity and inducing apoptosis. Additionally, ROS can deactivate antioxidant defense systems, including SOD and GSH. ROS attacks unsaturated fatty acids in the plasma membrane, initiating membrane lipid peroxidation and ultimately leading to the production of MDA. SOD serves as a crucial enzyme in protecting the body from oxidative stress [32,33], while GSH acts as a potent intracellular antioxidant, facilitating the decomposition of hydrogen peroxide and inhibiting the generation of free radicals. The accumulation of neutrophils represents a key mechanism underlying the progression of ALI. MPO activity serves as a marker of neutrophil activation and aggregation in the inflammatory response [32]. In our investigation, following RRFJ intervention, lung tissue exhibited increased SOD and GSH activity alongside reduced MPO and MDA levels, indicating RRFJ’s ability to alleviate oxidative stress and reduce neutrophil aggregation within lung tissue.

In metabolomics studies, blood and lung tissue are chosen as ideal specimens for metabolic analysis. From a physiological standpoint, the lungs are the primary organs affected by disease, while blood contains a diverse range of small molecular metabolites. Physiologically, the lungs are the main organs impacted by sickness, while the blood includes a wide variety of tiny molecular compounds. Alterations in blood composition can indicate both normal and abnormal changes in organs. During illness, inflammatory mediators present in the bloodstream can migrate to the lungs and initiate tissue damage. In this experiment, we utilized UPLC-QTOF-MS/MS to characterize and analyze endogenous metabolites in the LPS-induced ALI mouse model treated with RRFJ. We identified 22 candidate metabolites in plasma and 24 candidate metabolites in lung tissue. These metabolites are mainly associated with pathways such as phenylalanine, tyrosine, and tryptophan biosynthesis; glyoxylate and dicarboxylate metabolism; Phenylalanine metabolism; arachidonic acid metabolism; tryptophan metabolism; alanine, aspartate, and glutamate metabolism; and tyrosine metabolism.

Glyoxylic acid acts as an intermediate in the conversion from isocitric acid to malic acid, playing a central role in the citric acid cycle and serving as a primary source of human energy. On the other hand, glyceric acid can be converted by glycate 3-kinase into 3-phosphoglycerate, which enters the glycolytic cycle to regulate energy metabolism [34]. Following RRFJ intervention, glyceric acid exhibited upregulation, while glyoxylic acid showed downregulation compared to the model group. This suggests that RRFJ enhances the conversion of glyoxylic acid to isocitric acid and increases glycerol content, thereby regulating energy metabolism to mitigate LPS-induced acute lung injury.

Amino acid metabolism emerges as another pathway through which RRFJ inhibits ALI. Both phenylalanine and tryptophan belong to aromatic amino acids. Phenylalanine, an essential amino acid, can exacerbate inflammation and immune responses. Recent findings correlate phenylalanine accumulation with COVID-19 markers, as its increase prompts alveolar epithelial cell protrusion, leading to inflammation and, ultimately, ARDS-related fatalities [35]. L-kynurenine and 5-hydroxytryptophan are intermediate products in tryptophan metabolism [36]. Research indicates elevated kynurenine levels in COVID-19 patients, which significantly decrease upon recovery [37]. Moreover, 5-hydroxytryptophan demonstrates anti-inflammatory properties in lung tissues [38]. Intriguingly, our study reveals decreased levels of both phenylalanine and kynurenine, along with increased 5-hydroxytryptophan levels following RRFJ intervention. This suggests RRFJ’s capability to modulate phenylalanine and tryptophan metabolism, thereby alleviating lung injury. In alanine, aspartate, and glutamate metabolism, aspartate can convert into glutamine, which in turn reduces IL-1β and TNF-α levels in plasma and tissues, thus ameliorating inflammation. Additionally, glutamine storage bolsters the body’s antioxidant capacity, mitigating oxidative stress [39,40]. In comparison to the model group, the RRFJ group exhibits increased levels of glutamine, glycine, and aspartic acid. This indicates RRFJ’s ability to regulate their metabolism, thereby mitigating inflammation and oxidative stress, ultimately alleviating LPS-induced acute lung injury.

Arachidonic acid (AA), an essential polyunsaturated fatty acid, is widely distributed throughout the human body, primarily bound to hydroxyl groups on glycerophospholipids. During oxidative stress, enhanced phospholipase A2 activity prompts AA release, primarily for synthesizing pro-inflammatory factors. These factors activate intracellular inflammatory signal transduction, amplifying inflammation by promoting the synthesis and release of cytokines such as TNF-α. The metabolism of AA via the cyclooxygenase (COX) and lipoxygenase (LOX) pathways generates reactive oxygen species, exacerbating oxidative stress. Prostaglandin E2, a key AA metabolite, enhances inflammation by regulating immune cell differentiation and cytokine expression. Additionally, 20-HETE, a product of CYP450-catalyzed AA metabolism, modulates eNOS activity, promoting pulmonary artery dilation by increasing nitric oxide release [41]. Our findings indicate significantly reduced AA levels in LPS-induced ALI mice following RRFJ intervention, suggesting RRFJ’s capacity to regulate AA metabolism and alleviate acute lung injury.

Through the analysis of the identification and network pharmacology of 44 potential active ingredients, we have found several important compounds and proteins in RRFJ’s anti-ALI effects. The top ten components, based on Degree value, include catechin, 4-coumaric acid, ellagic acid, abscisic acid, chrysin, emodin, gallic acid, 4-hydroxybenzaldehyde, ethyl gallate, and quercetin. Among these, 4-coumaric acid exhibits notable anti-inflammatory and antioxidant effects, reducing levels of TNF-α, IL-1β, and myeloperoxidase (MPO) through the Nrf2 signaling pathway [42]. Quercetin inhibits TNF-α production in macrophages induced by LPS, as well as IL-8 production in lung A549 cells [43]. Gallic acid’s anti-inflammatory mechanism involves the MAPK and NF-κB signaling pathways, mitigating inflammation by suppressing the release of inflammatory mediators and cell infiltration [40]. Ellagic acid, a natural phenolic compound, exerts potent antioxidant and free-radical-scavenging activities, alleviating oxidative stress [44,45]. In our study, we identified 40 core genes primarily involved in apoptosis and inflammatory responses. AKT1, a key regulatory kinase in the PI3K/AKT signaling pathway, controls cell proliferation, survival, and the cell cycle as the primary downstream target protein for PI3K. Additionally, phosphorylated AKT1 regulates downstream factors in inflammatory responses [46]. HSP90AA1 is closely associated with macrophages, which may contribute to the production of effective cytokines such as TNF-α and IL-6, thereby enhancing T lymphocytes, neutrophils, and other immune cells that help enhance immunity [47]. Caspase-3, pivotal in cell apoptosis, reverses pulmonary endothelial cell apoptosis, mitigating LPS-induced ARDS and lung injury. It acts as a nexus between endogenous and exogenous apoptotic pathways, functioning prominently as a key apoptotic protein. Its expression level correlates directly with the occurrence of cell apoptosis, with studies demonstrating elevated Caspase-3 expression in LPS-induced acute lung injury [31].

By integrating the results of network pharmacology and metabolomics, we identified eight key targets (HAO1, MPO, PTGS2, ALOX5, CYP2D6, CYP2C9, CYP3A4, TH, and KMO) and five related metabolic pathways (arachidonic acid metabolism, tryptophan metabolism, glyoxylate and dicarboxylic acid metabolism, phenylalanine metabolism, and tyrosine metabolism). ALOX5 serves as a crucial rate-limiting enzyme in leukotriene synthesis, playing a pivotal role in inflammation mediation and the onset of diverse human diseases. It is well established that ALOX5 activation instigates inflammatory responses and initiates multiple cell death pathways, including apoptosis, ferroptosis, and pyroptosis [48]. MPO facilitates oxidative stress by stimulating the generation of reactive oxygen species and reactive nitrogen species. It also modulates the polarization and signaling pathways associated with inflammation in microglia and neutrophils [49]. PTGS2, also known as cyclooxygenase, catalyzes the conversion of arachidonic acid (AA) into prostaglandins (PGs), which are primarily implicated in inflammation, cell proliferation, and apoptosis [50]. CYP2D6, CYP2C9, and CYP3A4 belong to the cytochrome P450 family [51]. Under normal physiological conditions, the kynurenine pathway plays a vital role in cellular energy generation and tryptophan catabolism. However, during inflammatory states, there is an upregulation of KP enzymes, particularly kynurenine 3-monooxygenase (KMO) [52]. 

Afterward, we proceeded with molecular docking analysis between the top 10 compounds in RRFJ, as determined by their degree values from network pharmacology results and these ten key proteins. Docking scores reflect the binding affinity of each compound with the target protein, with lower scores indicating tighter binding. The results show that most compounds exhibit high binding affinity across various proteins, particularly gallic acid, ellagic acid, and quercetin, which demonstrate favorable binding performance across multiple proteins. Quercetin displays low docking scores in most proteins, especially showing optimal performance in CYP2D6 and MPO, suggesting its high binding affinity with these proteins. Conversely, CYP3A4 and TH exhibit weaker binding affinity with quercetin, indicating its poorer binding performance with these proteins. Furthermore, emodin shows higher docking scores in CYP2E1 and TH, while its binding affinity with other proteins is relatively weak, suggesting better binding performance with CYP2E1 and TH but poorer binding performance with other proteins.

## 5. Conclusions

In this study, we employed a comprehensive approach integrating metabolomics, network pharmacology, and molecular docking to investigate the mechanism underlying RRFJ’s alleviation of LPS-induced ALI. Our findings demonstrate that RRFJ exerts a significant protective effect on ALI mice by reducing levels of inflammatory cytokines (TNF-α, IL-6, IL-1β), enhancing antioxidant capacity through increased SOD activity and GSH content, and decreasing MDA content and MPO activity. Additionally, RRFJ indirectly alleviates LPS-induced ALI by modulating amino acid metabolism and arachidonic acid metabolism. The combined analysis of metabolomics and network pharmacology suggests that RRFJ treatment of ALI primarily involves improving inflammatory response and regulating cell apoptosis (Figure 7).

## Figures and Tables

**Figure 1 nutrients-16-01376-f001:**
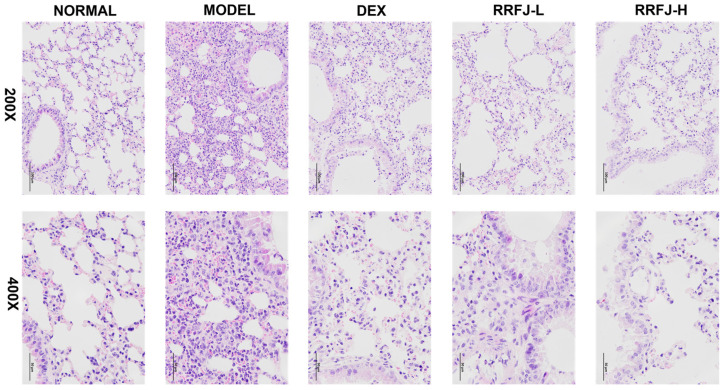
Effect of RRFJ on lung tissue in mice (H&E staining, original magnification ×200 and ×400).

**Figure 2 nutrients-16-01376-f002:**
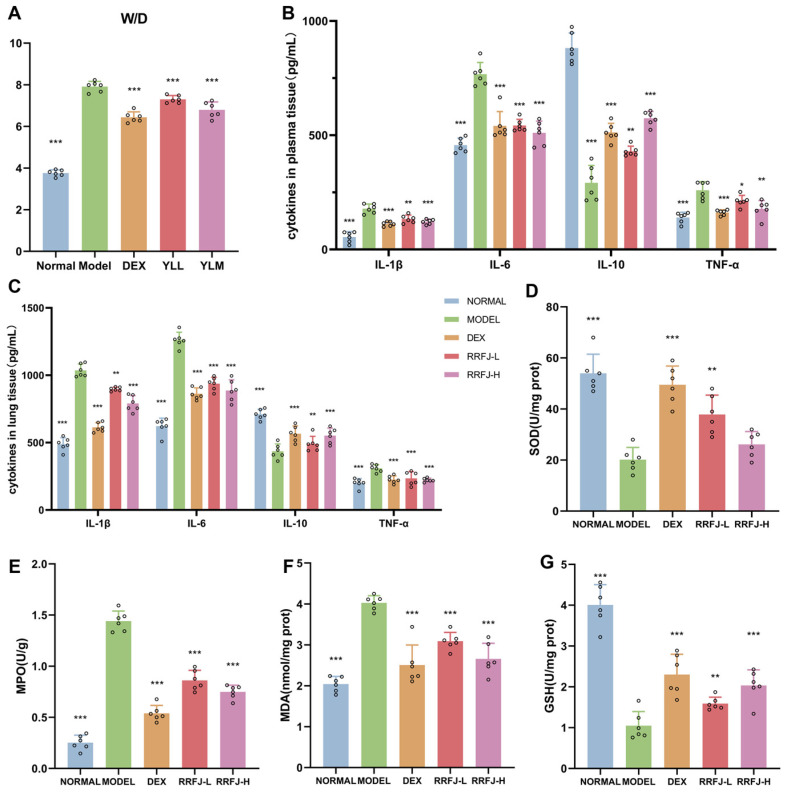
Effects of RRFJ on W/D ratio in lungs, systematic inflammatory response in plasma and lung tissue and oxidative stress in lung tissue in LPS-induced mice. (**A**) The W/D ratio in lungs (*n* = 6). (**B**) The systemic inflammatory response in plasma. (**C**) The systemic inflammatory response in lung tissue. (**D**–**G**) MPO activity, MDA content, SOD activity, and GSH content in lung tissue are present. Compared with the LPS group, * *p* < 0.05, ** *p* < 0.01, *** *p* < 0.001.

**Figure 3 nutrients-16-01376-f003:**
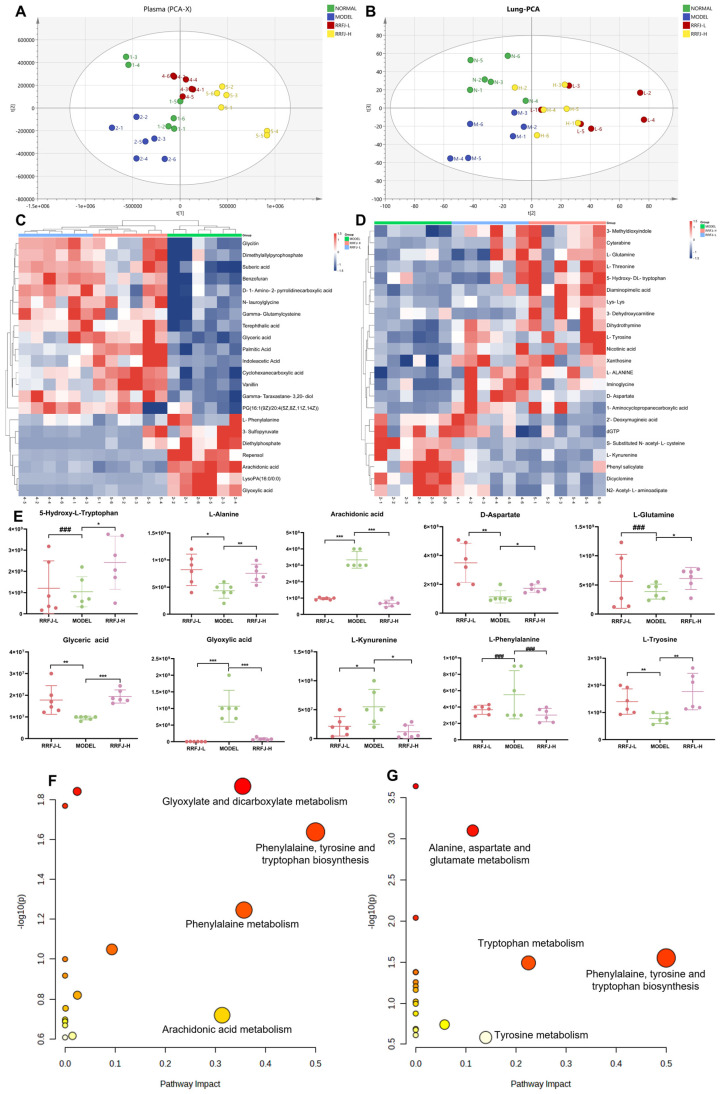
RRFJ modulated the plasma and lung metabolome: (**A**) PCA diagram of plasma; (**B**) PCA diagram of lung tissue; (**C**) heatmap of 22 differentially expressed metabolites in plasma; (**D**) heatmap of 24 differentially expressed metabolites in lung tissue; (**E**) change in the expression level of the 10 selected metabolites with a significant difference; (**F**) enrichment analysis of differentially expressed metabolites in plasma; (**G**) enrichment analysis of differentially expressed metabolites in lung tissue (### = no statistics, * *p* < 0.05, ** < 0.01, *** *p* < 0.001).

**Figure 4 nutrients-16-01376-f004:**
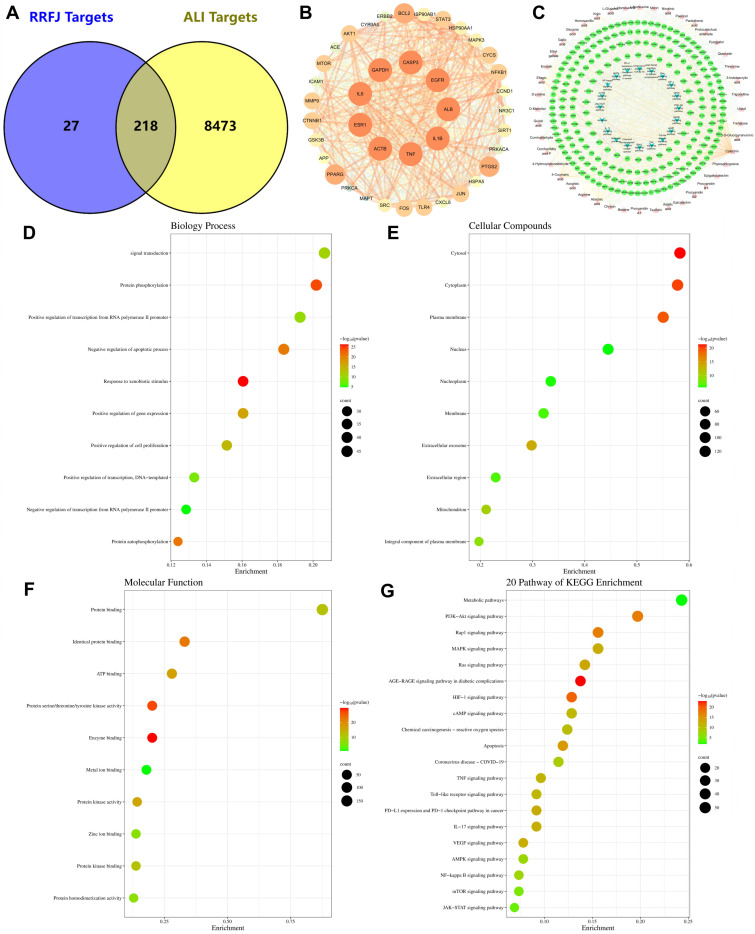
Network pharmacological analysis of RRFJ in the treatment of ALI. (**A**) Venn diagram of shared targets between RRFJ and ALI. (**B**) Core genes for RRFJ treatment of ALI. (**C**) “RRFJ active components-target-Pathway” network of anti-ALI activity of RRFJ. (**D**–**F**) GO enrichment analysis of potential targets for active compounds from RRFJ against ALI. BP, biological process; CC, cellular components; MF, molecular function. (**G**) KEGG enrichment analysis of potential targets for active compounds from RRFJ against ALI.

**Figure 5 nutrients-16-01376-f005:**
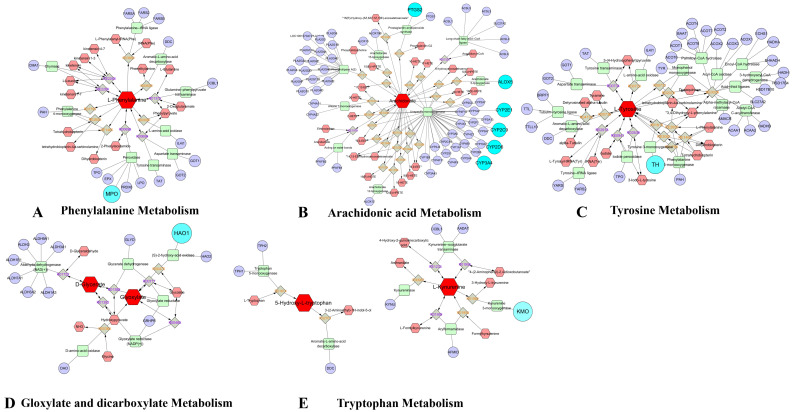
“Potential metabolite-reaction-enzyme-gene’’ interaction network. Red hexagons represent metabolites, gray diamonds represent reactions, green rectangles represent enzymes, and purple circles represent genes.

**Figure 6 nutrients-16-01376-f006:**
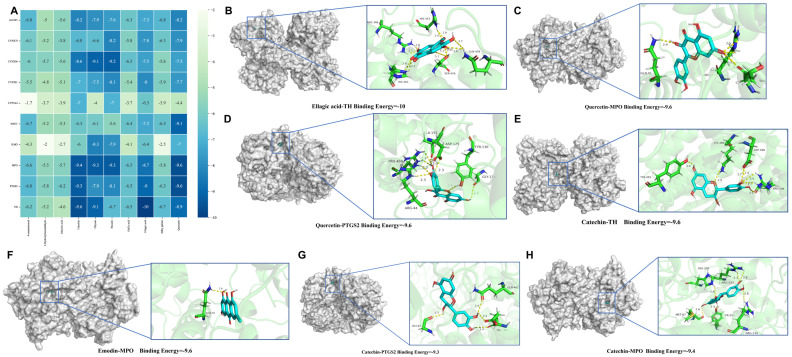
Molecular models of 10 RRFJ core active components binding to the 8 most promising target proteins on ALI. (**A**) Heat map of minimum binding energy. (**B**−**H**) Visualization of molecular docking with minimum binding energy less than −9 KJ/mol.

**Figure 7 nutrients-16-01376-f007:**
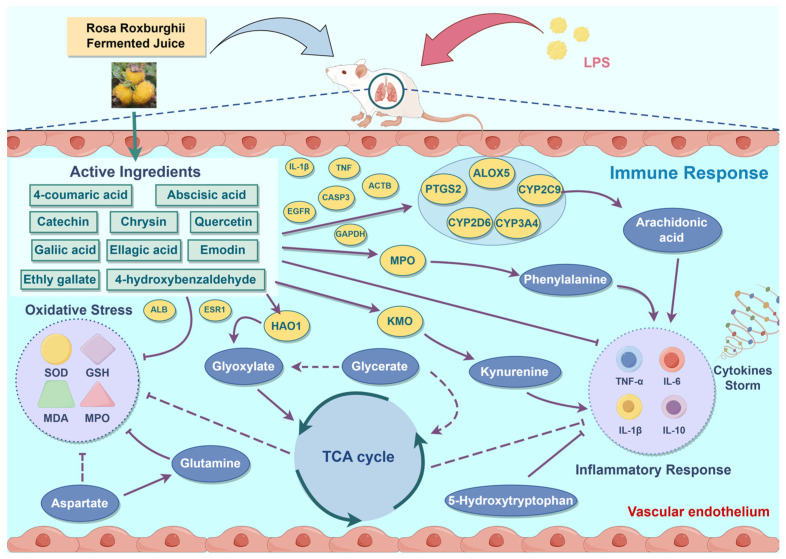
Schematic diagram representing the improvement mechanism of RRFJ in LPS-induced ALI mice (by Figdraw).

**Table 1 nutrients-16-01376-t001:** Tentative identification of potential active ingredients in RRFJ.

NO	Identification Name	Molecular Formular	RT (min)	Observed MS1 (*m*/*z*)	MS2 (*m*/*z*)
1	L-Threonine	C_4_H_9_NO_3_	0.923	120.0657 (+1.55)	102.05528, 84.04484, 74.06068, 56.05026
2	Gluconic acid	C_6_H_12_O_7_	0.944	195.0503 (−3.66)	159.02901, 129.01617, 75.06745
3	β-D-Glucopyranuronic acid	C_6_H_10_O_7_	0.946	193.0349 (−2.84)	113.02319, 103.00240, 85.02815, 72.99180
4	D-Mannitol	C_6_H_14_O_6_	0.947	181.6710 (+1.74)	163.06003, 131.03355, 119.03498, 101.02442, 89.0244, 71.01385
5	Betaine	C_15_H_11_NO_2_	0.965	118.0865 (+1.96)	59.07378
6	Trehalose	C_12_H_22_O_11_	0.989	341.1093 (−0.17)	179.05525, 119.03376, 89.02309, 71.01254
7	Quinic acid	C_7_H_12_O_6_	0.993	191.0192 (−3.20)	173.04477, 127.03871, 85.02814
8	Trigonelline	C_7_H_7_NO_2_	1.009	138.0549 (−0.17)	110.06035, 94.0656
9	L-Glutamic acid	C_5_H_9_NO_4_	1.016	148.0604 (+0.35)	130.0500, 102.05538, 84.04497, 56.03430
10	D-proline	C_5_H_9_NO_2_	1.049	124.0394 (+1.07)	96.04478, 80.05005
11	Arginine	C_6_H_14_N_4_O_2_	1.057	175.1078 (+0.00)	130.09749, 98.06047, 70.06585
12	Pantothenic acid	C_9_H_17_NO_5_	1.902	220.1180 (+1.92)	174.11255, 156.10155
13	L-Homoserine	C_4_H_9_NO_3_	1.118	120.0657 (+1.55)	102.05528, 84.04486, 74.06068, 56.05026
14	Pyrogallol	C_6_H_6_O_3_	1.235	127.0393 (+1.31)	109.02876, 81.03349, 53.03942
15	Ascorbic acid	C_6_H_14_O_6_	1.25	175.0583 (−1.65)	115.00425, 87.00745
16	Uracil	C_4_H_4_N_2_O	1.284	113.0239 (+4.08)	96.00851
17	Nicotinic acid	C_6_H_5_NO_2_	1.322	124.0394 (+1.07)	96.04478, 80.05005, 70.02950
18	Kojic acid	C_6_H_6_O_4_	1.697	143.0340 (−0.06)	125.02338, 113.02338, 97.02879, 69.03418
19	L-Norleucine	C_6_H_13_NO_2_	1.892	132.1022 (+0.99)	86.09695, 69.07059
20	Gallic acid	C_7_H_6_O_5_	2.074	169.0136 (−3.71)	125.02316, 79.01754
21	Epigallocatechin	C_15_H_14_O_7_	4.746	305.0672 (−0.29)	261.067605, 125.03291
22	3-Indoleacrylic acid	C_11_H_9_NO_2_	6.494	188.0705 (−0.31)	170.06004, 146.06003, 118.06513, 91.05470
23	Procyanidin A1	C_30_H_24_O_12_	6.878	575.1204 (+0.27)	423.07266
24	Procyanidin B2	C_3O_H_26_O_12_	6.921	579.1489 (+1.28)	427.10135, 409.09106, 291.0858
25	Procyanidin B1	C_30_H_26_O_12_	6.922	577.1354 (−0.92)	425.08820, 407.07663, 289.0720, 291.0871
26	Emodin	C_15_H_10_O_5_	6.957	271.0596 (−1.75)	229.04954, 197.05971, 173.05971, 145.06490, 131.04893
27	4-Hydroxybenzaldehyde	C_7_H_6_O_2_	7.144	123.0440 (−1.72)	95.04955, 77.03918
28	Catechin	C_15_H_14_O_6_	7.165	290.07458 (−1.06)	245.0448, 205.04993, 203.07065, 179.03412, 151.03908
29	Homovanillic acid	C_9_H_16_O_4_	7.268	181.0500 (−3.36)	166.02579, 137.05962, 122.03733, 107.04891
30	p-Coumaric acid	C_9_H_8_O_3_	7.307	165.0544 (−0.35)	147.04381, 119.04916, 91.05460, 65.03925
31	Epicatechin	C_15_H_14_O_6_	7.795	291.0858 (−2.46)	245.04413, 161.05949, 139.03885, 123.0446
32	Chrysin	C_15_H_14_O_7_	7.953	255.0643 (−3.30)	181.06543, 153.06988, 68.9910
33	Protocatechualdehyde	C_7_H_6_O_3_	7.99	139.0388 (−2.76)	111.04431, 93.03349, 83.04969
34	Cuminaldehyde	C_10_H_12_O	8.545	149.0961 (−0.46)	134.07307, 105.07024, 79.05423
35	Ellagic acid	C_7_H_6_O	8.815	301.0002 (+1.31)	283.99600, 257.00867, 229.01373
36	Ethyl gallate	C_9_H_10_O_5_	8.896	197.0449 (−2.62)	169.01338, 125.02332, 78.95766
37	Taxifolin	C_15_H_12_O_7_	9.27	303.0514 (+0.98)	285.04044, 259.06107, 153.08145, 125.02322
38	Paeonol	C_9_H_10_O_3_	10.593	167.0128 (−1.59)	149.96600, 121.06496, 84.96029
39	Abscisic acid	C_15_H_20_O_4_	11.015	263.1290 (+0.06)	245.13214, 204.11478, 161.09610, 111.04515
40	Quercetin	C_15_H_10_O_7_	11.945	301.0358 (+0.51)	151.00368, 121.02823
41	Morin	C_15_H_20_O_7_	11.924	302.0425 (+0.51)	229.01344, 151.00255
42	Asiatic acid	C_36_H_48_O_5_	13.222	471.3464 (−2.13)	453.33633, 407.33084, 203.17943, 107.08553
43	Corchorifatty acid F	C_18_H_32_O_5_	13.556	327.2178 (+0.79)	229.14453, 209.11832, 185.11831, 127.11284, 97.06589, 85.02950
44	Phytosphingosine	C_18_H_39_NO_3_	16.997	318.2997 (−2.22)	300.28851, 256.26318, 85.10162

## Data Availability

The data presented in this study are available on request from the corresponding author due to privacy.

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
