# Peer review of "Analysis of the Protective Effects of Rosa roxburghii-Fermented Juice on Lipopolysaccharide-Induced Acute Lung Injury in Mice through Network Pharmacology and Metabolomics"

_nutrients, 2024, doi:10.3390/nu16091376_

Round 1
Reviewer 1 Report
Comments and Suggestions for Authors
1. Line number 189-190, the sentence “The obtained compound structure was identified as potential targets for RRTP” is not clear to me. please rewrite.
2. In this study the bioinformatics analysis is not that necessary and logical, and the results are not clearly explained.
3. The figures included in the manuscript are not clearly visible, which hampers the ability to discern the results effectively. Please provide high-resolution images to enhance the clarity and utility of the visual data
4. Section 3.9, line number 373-374, “If the binding energy… with the target protein” is there any reference for this statement? The docking analysis would benefit from a comparison with a positive control to establish a baseline for interpretation. The statement, "Visualize docking results with molecular binding energy less than -9 KJ/mol," should be revised for clarity and accuracy. “These targets include the removal and hydrogenation of ethanol and water, as well as the optimization and repair of amino acids.” Consider revising this sentence. Please review and refine this section comprehensively to improve accuracy and coherence.
5. The manuscript contains several typographical (for example; 'tthe' in the abstract (line number 20))and grammatical errors that could detract from its professional quality. It is recommended that the manuscript undergo thorough proofreading by a native English speaker or professional editing service to correct these issues. Additionally, some sentences are overly complex or awkwardly structured, which could impede understanding. Simplifying and clarifying these sentences would greatly enhance readability.
Reviewer 2 Report
Comments and Suggestions for Authors
The scientific botanical names should be given in italics but the author's name should not be. The species name should not be capitalized. So, the right notation for the title species is Rosa roxburghii Tratt. Usually, the abbreviation is given as R. roxburghii. The corrections concerning this comment should be done throughout the text.
Authors should avoid redundant information. In the “2. Materials and Methods” section: the sub-sections “2.2 UHPLC-ESI-Q-Exactive Plus Orbitrap-MS Analysis” and “2.9 Metabolomics Analysis” gave similar information. These sub-sections should be re-designed to avoid the redundant information.
In Table 1: the columns should be given in the following order (from left to right): No; Metabolite name; RT; MS1 in m/z (∆ppm); MS2 in m/z (relative abundance). The authors should establish the quantities of each metabolite in the fermented juice. In section “3. Results”, “3.1 Tentative Identification of RRFJ Active Ingredients.”: A brief discussion on the identification of each metabolite in RRFJ should be given as well as citing an appropriate reference. The authors should state which metabolite was reported here for the first time.
